Emerging IoT domains, current standings and open research challenges: a review

http://orcid.org/0000-0002-0114-0457 Ali Omer 1 2
http://orcid.org/0000-0002-3554-0061 Ishak Mohamad Khairi 1 khairiishak@usm.my
Bhatti Muhammad Kamran Liaquat 2
1 School of Electrical and Electronic Engineering, Universiti Sains Malaysia (USM) , Nibong Tebal, Pulau Pinang , Malaysia
2 Department of Electrical Engineering, NFC Institute of Engineering and Technology (NFC IET) Multan, Punjab , Pakistan
Shang Yilun
Electronic publication date: 2021 Aug 16
Publication date: 2021
Volume: 7
Electronic Location ID: e659
Received 2021 Apr 13; Accepted 2021 Jul 13
Copyright: © 2021 Ali et al.
Copyright year: 2021
Copyright holder: Ali et al.
License: This is an open access article distributed under the terms of the Creative Commons Attribution License, which permits unrestricted use, distribution, reproduction and adaptation in any medium and for any purpose provided that it is properly attributed. For attribution, the original author(s), title, publication source (PeerJ Computer Science) and either DOI or URL of the article must be cited.
License URL: https://creativecommons.org/licenses/by/4.0/

Keywords: Internet of things, Internet of nanothings, Internet of spacethings, Internet of underwater things, Underwater communications, Social internet of things

Funding: Universiti Sains Malaysia (USM) Research Grant RUI:1001/PELECT/8014049 This work was sponsored by a Universiti Sains Malaysia (USM) Research Grant (RUI:1001/PELECT/8014049). The funders had no role in study design, data collection and analysis, decision to publish, or preparation of the manuscript.

==============================
Over the last decade, the Internet of Things (IoT) domain has grown dramatically, from ultra-low-power hardware design to cloud-based solutions, and now, with the rise of 5G technology, a new horizon for edge computing on IoT devices will be introduced. A wide range of communication technologies has steadily evolved in recent years, representing a diverse range of domain areas and communication specifications. Because of the heterogeneity of technology and interconnectivity, the true realisation of the IoT ecosystem is currently hampered by multiple dynamic integration challenges. In this context, several emerging IoT domains necessitate a complete re-modeling, design, and standardisation from the ground up in order to achieve seamless IoT ecosystem integration. The Internet of Nano-Things (IoNT), Internet of Space-Things (IoST), Internet of Underwater-Things (IoUT) and Social Internet of Things (SIoT) are investigated in this paper with a broad future scope based on their integration and ability to source other IoT domains by highlighting their application domains, state-of-the-art research, and open challenges. To the best of our knowledge, there is little or no information on the current state of these ecosystems, which is the motivating factor behind this article. Finally, the paper summarises the integration of these ecosystems with current IoT domains and suggests future directions for overcoming the challenges.

Introduction

The rapid deployment of the Internet of Things (IoT) and its adaptation in domestic, industrial, and military domains fueled research into previously unknown ecosystems. This necessitates the real-time integration of processes and data, which provides automated actionable insights for intelligent future applications. IoT systems are mostly considered a hybrid integration of technology that integrates with various digital domains (Accenture, 2016; Hermann, Pentek & Otto, 2016; Al-Fuqaha et al., 2015; Khan et al., 2012). In contrast to early developments in this industry, the current digital devices witnessed an increase in device computing power, reduced physical device size and power, as well as a significant reduction in capital cost. Due to these contributing factors, IoT technology is making waves in various digital domains; and opens possibilities to emerging IoT ecosystems.

It is estimated to have over 75 billion IoT devices by the year 2025 (Department, 2018). Since Gordon Moore’s forecast of transistor density doubling every year, device performance and capabilities have grown steadily and tremendously (Emer, 2008). Another interesting viewpoint for IoT enabled domains is the device and application heterogeneity. According to Nokia, an IoT horizontal platform with standardized protocols and open interfaces will help network values grow exponentially and allow developers to focus on the real differentiators to architect fast scaling vertical solutions. However, the more choices there are in terms of technologies to deploy these applications, the more heterogeneity is observed that hampers the effective deployment of IoT technology. From a developer’s viewpoint, the vertical market solutions also face the same challenges with the availability of a large number of operating systems, middle-wares and enterprise solutions (Wollschlaeger, Sauter & Jasperneite, 2017; Silva, Khan & Han, 2018).

The current IoT landscape suffers greatly from the technological heterogeneity, protocols, and lack of standardization (Silva, Khan & Han, 2018). In addition, enterprise solutions offer a minimal interoperability, thus tightly coupling consumers, IoT domains and applications to their own ecosystems. In this regard, open-source development including firmware, middlewares and platforms are mostly deployed to overcome these limiting factors. It is also very important to consider that most of the IoT application deployments rely greatly on the common technologies in IoT architecture layers. However, integration of these technologies requires standardization at technology, as well as at application layers. Therefore, it is fundamentally important to consider IoT applications in terms of integrated ecosystems where machine to machine (M2M) communication, business to business (B2B) processes and applications are flexible in both scalability and integration (Mohammadi et al., 2018; Jazdi, 2014). The IoT ecosystem map as given by Fig. 1, explains this phenomenon granularly, where several policy roles and processes are aligned at every layer of operation.

Figure 1 IoT ecosystems map and open challenges.

In addition to the focus on the aforementioned IoT domains, the emerging IoT ecosystems promise enormous technological possibilities that could have a direct quality of experience (QoE) impact on humans. For instance, Internet of SpaceThings (IoST) envisions high speed, low-latency, umbrella internet coverage to all parts of the world. Internet of NanoThings (IoNT) may facilitate telemedicine, inter wireless body area network (WBAN) communications that can revolutionize the healthcare industry. Internet of UnderwaterThings (IoUT) can help to improve the quality of our oceans, speed up search and rescue operations, as well as enable reliable disaster management systems (such as Tsunami and oil leakage alerts) to save previous humans lives. On the other hand, the Social Internet of Things (SIoT) envisions interfacing IoT networks to humans and social networks (SNs) that can bring near realtime intelligible insights. However, despite the technological advancements and abundance of technical platforms, these IoT ecosystems require a ground-up technological re-modelling. These emerging ecosystems have distinct operational conditions that require advanced research and design in the IoT technology stack. In addition, ecosystems such as SIoT require comprehensive security, privacy, and governance policies. These emerging ecosystems not only require technological advancements but, at the same time, a unified scalable architecture. Therefore, it is important to investigate the open research challenges in these domains individually.

In this paper, an extensive review is carried out on several emerging IoT ecosystems with a vast future scope based on their integration and ability to source other IoT domains. This paper provides a detailed overview of these ecosystems by highlighting their application domains, state-of-the-art research, and open challenges. Some of the major contributions of this paper can be concluded as:Identification of emerging IoT domains and ecosystems.

Detailed investigation on the state-of-the-art IoT ecosystems and associated technologies.

Application-specific deployment examples and trends in emerging IoT ecosystems.

Security, privacy, and trust requirements for SIoT.

In the following sections, each of these ecosystems is investigated by focusing on the technological requirements and current research trends. Finally, the paper concludes by providing future directions to overcome these challenges that could enable swift development and integration of these ecosystems with current IoT domains.

Research design

This article’s primary research goals are twofold. Firstly, this article attempts to classify rapidly evolving IoT domains which can be considered an independent ecosystem. In order to classify these IoT application domains, a thorough analysis of journal papers, scientific reviews and news was conducted. Second, this article investigates the architectural, design, technological, and integration challenges that exist in these domains. A rigorous systematic literature review (SLR) was conducted to this end. SLR facilitates in-depth systematic study of a topic by assessing emerging literature, cutting-edge technology advances, and academic gaps in these fields. This enabled us to examine potential future research directions in these settings.

Research questions

We surveyed real-world IoT technology domains in the preliminary research phase to identify gaps in emerging technologies. Towards this end, we discovered a multitude of surveys and research papers on the most common IoT application domains. Among these application areas are the Industrial Internet of Things (IIoT), IoT for wearable devices, renewable applications, and smart cities, among others. The preliminary survey assisted in contrasting these prominent application domains with fast-emerging, under-reported domains. Furthermore, we found a small number of surveys that specifically report these recent trends, which served as the impetus for this research.

Besides that, we have developed research questions, as proposed by Gough, Oliver & Thomas (2017) and Boland, Cherry & Dickson (2017) to find research gaps in these domains. This prompted the formulation of (RQ1): “What are the rapidly evolving IoT domains that might constitute an independent ecosystem?”. This motivated us to expand our research into (RQ2): “How do the architectural and technical requirements in these IoT domains differ from those in existing application domains?”. In the following steps, we explored the integration of these ecosystems with the existing IoT landscape. This naturally led to (RQ3): “How can these ecosystems be integrated into the existing digital blueprint?”. Finally, if these ecosystems are adapted using existing solutions, what are the technological problems that must be solved, that are formulated in terms of (RQ4): “What open research challenges exist in this domain?”.

As a result, this review article is organized around the following research questions:What are the rapidly evolving IoT domains that might constitute an independent ecosystem? This question aims to identify the rapidly evolving application specific IoT domains.

How do the architectural and technical requirements in these IoT domains differ from those in existing application domains? This question allowed us to classify the differences in technological requirements for these IoT domains.

How can these ecosystems be integrated into the existing digital blueprint? The integration effect, feasibility, and scope of these IoT application domains are the focus of this issue. It also allows for the investigation of technical specifications, mostly at the communication layer, in order to seamlessly integrate with current IoT applications.

What open research challenges exist in this domain?. The majority of current IoT communication relies on wireless connectivity. The IoT domains presented in this article, however, do not use this connectivity option. This question aims to clarify the most pressing technological challenges in these areas.

Search criteria

Initially, a broad search was conducted to classify the full range of IoT technology domains. This allowed us to identify the most recent literature on some of the most significant IoT domains. Following that, we conducted a depth-wise search to investigate state-of-the-art research in gap areas. Furthermore, based on our formulated research questions, composite search strings were implied to study the research topics.

The following search terms were used:

“Nano things” OR “nano communications”, OR “IoT Space” OR “Space communications”, OR “Underwater communications”, OR “crowdsourcing model” OR “Social IoT”. The search strings allowed for depth-wise exploration of various emerging IoT ecosystems. To that end, we suggested categorical application domains in our study. This will help readers classify and categorise the literature for these emerging domains. These search strings were applied to domain-specific indexed scientific databases (including Web of Science (WoS) and Scopus). Finally, Google Scholar and the Google search engine were used to discover the most recent technological reports on particular questions. While technical reports were used for implementation examples, peer-reviewed journals were preferred for reporting state-of-the-art analysis in their respective categories.

Scope of the review

Particularly, the search strings yielded a diverse set of topics in these IoT domains. However, we primarily covered the proposed architecture, communications, and integration aspects of these domains. Furthermore, several papers were referenced to define the fundamental building blocks in these domains. Finally, some of the surveys in these fields were consulted in order to comprehend the general technical problems in these areas.

Emerging domains: technological advancements and open challenges

After decades of IoT’s conception, a wide array of communication technologies progressively evolved in recent years which represent a variety of applications and communication requirements. The heterogeneity and fragmentation of the connective environment are now preventing the IoT concept from being fully realised by presenting various significant integration problems (Palattella et al., 2016). The ultimate objective of IoT is the introduction of plug and play technology that provides end users with easy operation, remote access control and settings. In this respect, 5G cellular networks are believed to be a potentially important driver for the global IoT which has yet to emerge with a connecting technology that is at once truly pervasive, reliable, scalable and cost-efficient (Shafique et al., 2020; Akpakwu et al., 2017).

As the next phase of the Internet of Things (IoT), this evolving integration of IoT and 5G will create numerous possibilities for a wide variety of applications, from telecommunications to automotive safety and e-healthcare for addressing human society’s complex problems (Sharma et al., 2020). The emerging IoT ecosystems discussed in this article may benefit from the 5G integration that could provide ultra-low latency, high bandwidth, and reliable backhaul networks (Palattella et al., 2016; Jaber et al., 2016). Although, this integration envisions a significant improvement in cycle of innovation, the interfacing challenges and technological complexities are yet to be addressed. One aspect of looking at these challenges is from a business perspective.

A lot of research has been conducted to study the impact of 5G, and human-in-the-loop for future IoT ecosystems. Mazhelis, Luoma & Warma (2012) defined the IoT Ecosystem from a commercial standpoint. Their study examined the basic technologies at each tier in order to create a generalised architecture that can accommodate vertical market solutions. Many researchers, on the other hand, foresee IoT Ecosystems that are linked to enterprise models. Similarly, Lee (2019) suggested a model that is inextricably linked to the enterprise business model. The subtle emphasis in this strategy is on enterprise-based IoT development rather than the underlying technologies. It has been proposed that developers, including hardware platform, application solutions, network technology, and software platform designers, collaborate to create an application-tailored IoT platform.

Although several enterprise frameworks for future IoT networks have been proposed, an uniform framework has yet to be established. To that end, it is critical to study the technological constraints and open research issues in each specific IoT ecosystem. As a result, this article investigates these emerging IoT ecosystems by examining the technological stack and the possibility of cross-technology integration in order to propose future implementation techniques.

Internet of Nano-things (IoNT)

The Internet of Nanothings (IoNT) might be viewed as a fusion of IoT with nanotechnology. In its most basic form, IoNT appears as a scaled-down application extension of IoT networks. The use of nanoscale sensors and ultra-small scale devices for sensing and actuation, on the other hand, is what sets this apart. IoNT offers tremendous development potential, particularly in the fields of environmental monitoring and medicinal applications.

Much like the conventional IoT networks, the aim of IoNT components is to sense information and communicate over longer distances. However, IoNT based communication is completely different as compared to traditional IoT networks. Data transmission in IoNT systems are mostly conducted utilizing either electromagnetic waves or via molecular communications as illustrated in Fig. 2. Given the application scenario, particularly the medical applications, IoNT sensors are low power devices and can only communicate over short distances using gateways and WBANs. At the heart of IoNT components, Nano nodes (NN) can be thought of sensors, that perform sensing and actuation. The NN can form multi-hop networks to communicate with each other, as well as gateway devices to transmit information. Therefore, it is important to investigate the technological advancements, communication techniques, and open challenges for these fundamental IoNT building blocks.

Figure 2 Communication options in wireless nano sensor networks.

IoNT architecture

With the advent of technology, the ever-increasing density of transistors on a chip, and the recent trend towards having a complete System-on-a-Chip (SoC), have seen a significant reduction in the size of electronic components and embedded systems. During early 1990s, many significant research players shifted their focus towards miniature electronic components, a design ambition that could allow the fabricating nanoscale devices. The first concept of nanoscale technology was SmartDust, presented to DARPA in early 1997 (Kahn, Katz & Pister, 2000). Initially, SmartDust was proposed to operate on radio frequency identification. However, it soon was practiced in full-swing in sensor networks to achieve sensor devices at a micrometer level (Kahn, Katz & Pister, 2000). These nanodevices went through different transformations, from design to communication challenges, power consumption, and susceptibility to the harsh environment (Warneke et al., 2001; Cook, Lanzisera & Pister, 2006).

Figure 3 indicates an IoNT network architecture where onboard sensors and intermolecular communications relay the external gateway information. Atakan, Galmes & Akan (2012) also presented a similar model based on molecular array communication. This novel network paradigm based on nanodevice intercommunication necessitates a number of components to create an IoNT architectural model. Some of these components are crucial IoNT architecture design, where their significance is given as:

Figure 3 Illustration of Internet of Nano-things (IoNT) communication architecture.

Nanonetworks of nano-nodes

They are the smallest and perhaps most basic nanomachines, performing functions such as computing and transmission. As a result, the biological nano-sensors inside human bodies and nanomachines incorporated into various objects provide the communication backbone of IoNT networks.

On/In-body microgateways as nano routers

They serve as information aggregators for nano-nodes. The behaviour of nano-nodes may be controlled by nanorouters using a basic control mechanism that includes device switching, power regulation, and duty cycle.

External gateways

It allows for remote control of the entire nanothings network over the Internet. In an intrabody network, for example, a smartphone can periodically or in the case of a specific activity communicate the information it gets from a nano-device put on our wrist to a doctor.

Nanocommunication in wireless body area networks (WBAN) mostly rely on inter-molecular communications. The preferred communication technology in Bio-Nano networks is mostly electromagentic waves. These resources and energy limitations are some of the most significant limiting factors for current research on the Internet of Bio-NanoThings (IoBNT). Dinc & Akan (2017), presented a framework to estimate Inter Symbol Interference (ISI) and Multi-User Interference (MUI) in a molecular communication network channel. The research also examined the safe threshold and transmitters to realize molecular communication for IoBNT. According to the recent literature, IoNT may be thought of two important domains. It is important to understand the building blocks of these domains, and the associated challenges, and research respectively.

IoNT domains

IoNT is divided into two domains: the Internet of Nano-Things Multimedia (IoMNT) and the Internet of Bio-Nano Things (IoBNT). The architecture of the nanodevices is dependent on the capabilities provided by nanotechnology (El-Din & Manjaiah, 2017). According to Jornet & Akyildiz (2012), IoMNT is defined as the connectivity of pervasively deployed multimedia nano-devices with existing communication networks and eventually the Internet establishes a novel communication paradigm that is further referred to as the Internet of Multimedia Nano-Things. According to this concept, nanocomponents must be combined into a single device that can execute common tasks, as illustrated by Fig. 4. The combination of System-on-Chip (SoC) with bio-sensors has the potential to provide a shared technical environment for the IoNT domains. As a result, it is critical to look at some of the common integrated nano components for both the IoBNT and IoMNT domains. Following that, certain architectural features of typical IoNT components may be defined as follows:

Figure 4 Conceptual architecture of multimedia nano-things (Jornet & Akyildiz, 2012).

Nano-sensors

The term "nanosensor" is not clearly established. Most definitions relate to a sensing device with at least one dimension less than 100 nm that collects information on the nanoscale and converts it into data for analysis. Nanosensors are chemical or mechanical sensors that can detect the presence of chemical compound changes, and in some cases monitor physical phenomenon such as like temperature. In addition, the nanotechnology deals with the chemical properties at nanoscale, thus aiding nanosensors to take advantages of these properties even while at micro-scale.

Nano-processors

Nanoprocessors are smaller, high-performance transistors that can operate at high frequencies. However, the intricacy of the processes that a nano-processor will be able to handle is determined by the number of transistors in the chip, and therefore by its overall size. Nanoscale processors are yet to surface, however, researchers have successfully demonstrated micro-scale carbon nanotubes (CNT) based processors that operates on CMOS logic (Sergiyenko, Molchanov & Orlova, 2019).

Nano-memories

True nanoscale memory banks are currently unavailable, owing mostly to their tiny size and difficulty to produce these components. However, as technology advances, researchers are exploring towards single-atom based nano-memories that may manage the most essential IoNT data in the near future.

Nano-transceivers

The use of nanomaterials has enabled various research to be conducted, as well as the potential of producing nano-antennas. These antennas are significantly smaller than typical antennas; they are made of graphene and can operate in the Terahertz frequency spectrum (Jornet & Akyildiz, 2014). The researchers are also investigating nanoscale transceiver design for IoNT domain that operates at higher frequencies (Shrivastava et al., 2020; Nafari & Jornet, 2017).

Nano-batteries

Unlike traditional energy storage systems, which use diverse battery chemistries to store energy, body area networks use chemical-free components to do otherwise. This demands the employment of electrical components capable of producing electricity via physiological changes such as temperature or pressure changes, or by using vibrational energy. Researchers are already investigating battery-less systems that could harvest energy by integrating piezoelectric components for energy generation (Cappelli et al., 2021).

The capabilities of nano-devices in terms of multimedia, processing, data storage, and energy may not always be the same; these capacities will vary depending on their size. Nanotechnologies are relatively new, developing rapidly, and have the potential to introduce cutting-edge technological capabilities in near future. As a result of its reliance on nanotechnology, the capabilities and potential of IoNT will expand. The communication aspect, on the other hand, is one of the major challenges in this domain. Next, these communication techniques and associated challenges are discussed.

Communication and energy conservation challenges in IoNT

To design the communication architecture in the IoNT domain, the researchers are focused on the high-frequency THz band. Due to their limited energy and processing capabilities, nano-sensors and nano-routers may not be able to function at such high frequencies directly; nevertheless, the usage of mico-gateways to compensate for data-rate and bandwidth is quite possible in future IoNT systems. In that regard, Jornet & Akyildiz (2011) investigated the Terra Hertz (THz) band for electromagnetic (EM) communication in these nanodevices. They also proposed graphene-based technology for nano-networks. In their research, channel capacity and interference were observed over a range of THz frequencies. It is suggested to be a suitable communication channel for nanoscale communication due to its very high channel capacity to support high data rates. However, THz frequency’s strong absorption dependency on vapour molecules severely attenuates the signals and introduces noise, thus making this frequency band very challenging for nano-networks communication. As the higher onboard power consumption may require a larger battery size, researchers investigate an alternate strategy to conserve and harvest energy for these nanodevices. In their research on energy harvesting in nanodevices, Jornet (2012) presented a highly opportunistic model based on nano piezo-electric-based energy generators working in THz frequency bands and accounts for temporal variations energy needs in the said band. The research provides a comprehensive energy consumption model in self-powered nanodevices working in nano-arrays, theoretically harvesting enough energy to transmit the packets with minimum loss. However, analytical models in this domain and a concrete proof-of-concept are yet to be achieved.

Another interesting application of IoNT may stem from plugging printed electronics circuits (Subramanian et al., 2005) to the internet. Manufacturers have been investing a great deal of effort in this technology, which may overcome some of the IoNT challenges. Chang, Facchetti & Reuss (2017) extensively reviewed the modern organic electronic printed technologies, which provide ample opportunities to utilize these brand-new solid-state fabrication techniques that we can utilize in IIoT and IoNT applications. These techniques can be implemented in health care, nano, and Industrial applications where onboard computing power and the real-estate of the device matter a lot.

In a similar vein, Ali & Abu-Elkheir (2015) presented a promising ubiquitous health care system model where IoNT devices form nano-network to run health care applications. Their research highlights some of the applications and architectural requirements needed for IoNT based health care applications. The transmission order of molecules provides independence and reduces the need for time synchronization to communicate. The proposed model MARCO was compared with several other molecular communication-based models providing higher communication rates. However, these communication rates are still in the order of Kilobits per second (Kbps). The lack of bandwidth and slower data rates opens research opportunities to investigate other communication technologies for higher throughput and low energy consumption. Miniaturization is a challenge in the IoNT domain, as there is always a trade-off between power consumption and the need for local processing capabilities. The node-level processing is directly proportional to the power consumption, where a high computational node will undoubtedly consume more power. Akyildiz & Jornet (2010) proposed different communication models for these nanoscale devices, especially in the health care industry where molecular communication and electromagnetic communications may appear as the only source of communication between these devices and a gateway device outside the human body.

Power consumption is a critical factor in all of these challenges. Nanodevices, in general, do not broadcast all of the time. To save energy, they only run one at a time and send only critical information based on aggregated events. Routing protocols are critical in designing the communication strategy that enhances network performance while conserving power. Balghusoon & Mahfoudh (2020) presented a state-of-the-art routing protocol for IoNT. The research provided insight into the WNSN and IoNT paradigms, as well as an examination of numerous contemporary routing protocols that are suitable for the features and functionality of nano-communication.

It is interesting to know that power consumption is always somehow translated in terms of battery cells’ real estate to provide power. The higher the power consumption, the larger the battery size, which further limits the actual fabrication of these nanodevices. Many studies have been conducted to wirelessly power these nanodevices as well as into energy harvesting capabilities to keep the device size minimal and enhance the lifetime (Movassaghi et al., 2014; Sarkar et al., 2014; Huang et al., 2012; Fotopoulou & Flynn, 2006). It is of paramount importance to balance how much processing capabilities and artificial intelligence are needed in the cloud or on the local node itself. These requirements tend to change abruptly from applications to application. A modular, self-adapting model is needed to realize these devices truly. In some recent research, alternate communication bands are rigorously investigated for nano-networks.

Finally, it is clear that traditional communication protocols may not be able to support nano-networks in terms of data throughput, scalability, and self-healing capacities. As a result, developing communication protocols for this domain is still considered a work in progress. Recent research in communication protocols, on the other hand, can be applied in the IoNT domain. In that regard, spectrum allocation, service discovery, channel sharing, intelligent routing and self-configurability are some of the most important open research challenges that are yet to be addressed.

Security and privacy challenges in IoNT

IoNT is being integrated into the majority of our daily lives, including health care applications, wearable sensors, and large-scale industrial infrastructures. These devices’ management and monitoring methods have been digitised and linked to the Internet, raising several security and privacy concerns. Because critical data is widely accessible over the Internet, hackers may quickly compromise it. Victims may suffer harm as a result of this, including theft, espionage, and data tampering. To secure sensitive data gathered by nanosensors, new security and privacy approaches are necessary. Dressler & Fischer (2015) believes that IoNT domain is prone to security attacks that can directly compromise not only the private data, but may also result in life-threatening situations by controlling the nanosensors such as implants, drug delivery, and wearable devices to name a few (Guo, Wei & Li, 2020). Some of the prominent security challenges and optimization goals for IoNT networks are discussed in Table 1.

Table 1 IoNT security challenges and optimization goals (Miraz et al. (2018)).

Challenges & goals	Resolution	
Data encryption	Most nanodevices frequently lack encryption owing to their small size and limited calculation functionality. If sensitive data between nanodevices cannot be encrypted, whether on a nanodevice alone or on nanonetworks, there are a number of safety problems especially when nanodevices become part of our bodies. Embedded cryptography, and lightweight genetic-based encryption schemes may help to encrypt IoNT data by keeping the energy footprint to minimum.	
Malware injection	Due to limited on-board processing capability, most nanosensors and wearable electronics interact via plain text with internal gateways. Furthermore, multi-hop wireless connection with external gateways expands the threat surface footprint, leading in man-in-the-middle and code injection attacks. These attacks have the potential to not only compromise private data, but also to convert these sensors into renegade devices. Time-access and token-based data relaying, in conjunction with identity management control methods, can aid in the detection of data breaches and code injection in real-time.	
Denial of service (DOS) attacks	DOS attacks have the potential to deplete network resources and traffic, resulting in a communication bottleneck. The first step in preventing DOS attacks is to analyse traffic and isolate rogue activity. To stop known attackers, the lightweight scalable DOS attack prevention method may be applied at external micro-gateways.	
Access control	The classic symmetric and asymmetrical cryptography usually ensures authentication. Biochemical cryptography is a new and uncharted topic that encrypts information and protects the secrecy and integrity of data by using biological molecules such as DNA/RNA evidence. Although this encryption technique opens up new application fields, it also raises new communication system problems. Recent research investigate DNA based cryptographic algorithm that can be implemented as user access control and authentication schemes (Ali et al., 2016; Popli, 2019; Niu et al., 2020).	

Security and privacy still stand as key open challenges for IoNT and wireless body area networks (WBAN). Healthcare data is vulnerable to passive attack and calls for stringent data protection and security procedures. The data must be protected as close to its origin so encryption systems are necessary for safeguarding applications focused on healthcare. To this end, several researchers have suggested lightweight encryption schemes to guarantee data protection and versatility for WBAN applications. The researchers in Jabeen et al. (2020) proposed a lightweight genetic-based data encryption technique that is bandwidth and power efficient. Similarly, a lightweight access control technology was proposed by Ullah et al. (2021), which focused on a signature-based access control system to ensure less computational cost, hence making it resource efficient.

IoNT ecosystem is vulnerable to a very large threat surface with multi-dimensional attack vectors. The current IoT standards are unable to address the dynamic security, privacy, and data regulation needs in IoNT domain, thus, requires a ground-up security control mechanism that is robust and scalable.

Remarks

The IoNT domain supports a wide range of application domains, including healthcare applications ranging from telemedicine to drug delivery. Furthermore, IoNT may be used in industrial applications such as temperature sensing, flu gas monitoring, structural analysis, and leakage detection, to mention a few. To prevent contamination and electromagnetic interference, IoNT electronics design necessitates well-packaged and insulated sensors. However, their practical limitations are owing to their small size, insulated packing, and vulnerability to electromagnetic interference.

Furthermore, IoNT domains have relatively low data rates, which limits their use in realtime applications. On the one hand, the researchers are investigating THz frequencies in order to circumvent bandwidth and data-rate limitations. Molecular communications, on the other hand, limit the use of high frequency applications owing to signal absorption and interference. Finally, the limited computing capabilities of NN hinders the usage of data encryption, posing a significant security risk for medical applications. Nanotechnology has gone a long way, but some of the most crucial hurdles that must be solved before it can reach its full potential include limiting data rates, energy harvesting, short-range communications, and data security.

Internet of Space-things (IoST)

The Internet of Space Things (IoST) is intended to achieve low-cost global networking that ensures high bandwidth umbrella network coverage. IoST is proposed as a solution to provide internet connectivity through satellite networks in Lower Earth Orbits (LEO) (Qu et al., 2017; Xia et al., 2019). IoT devices rely on traditional networks based on wired and wireless networks that are continuously evolving. One which is evolving and ever-increasing in parallel with the increase of these devices is the need for higher bandwidth and data rate. The densely intertwined internet is already experiencing addressing issues. In the future, with billions of these devices sending large streams of data around the clock, it will only get more congested. IoST is a proposed solution to carry this IoT traffic, which demands high data rates and throughput (Fraire, De Jonckère & Burleigh, 2021). It is imperative to know that satellite networks are thought of as alternative communication mediums to provide continuous network coverage, coverage in remote areas, and the backhaul communication network during disaster scenarios (Ban et al., 2020).

Large Enterprises such as Google (Kileon, 2018), Facebook (Statt, 2018), Space X (Foust, 2018; IEEE, 2019; Foust, 2017), and OneWeb (Clark, 2015) are already investing a great deal of both the revenue and the resources to provide internet to remote areas of the world. The aim is to build and push small-scale service delivery satellites in lower orbits to build a satellite network that can carry IoT traffic. SpaceX recently launched 10,000 satellites in (LEO) to build a large constellation of satellites (Foust, 2018; Ghafar, Castro & Khedr, 2019; McDowell, 2020). Starlink project is expected to launch over 30,000 TinTin satellites in the next decade to provide umbrella high-speed internet coverage to remote parts of the world. These satellites form constellations by using laser beams to communicate with each other (Liu et al., 2020; De la Osa et al., 2021). Other aspects that enable satellite networks as future communication link contenders are their ability to provide flexible bandwidth and very high data rates ranging up to Gigabits per second (Sansone et al., 2020). The flexible bandwidth will suit well for an on-demand service delivery model that can connect these devices to cover a broader area.

Deep space exploration is mostly favored at Higher Earth Orbits (HEO), whereas LEO are particularly preferred for application service domains, terrain monitoring and climate change monitoring (Lu et al., 2019). A detailed list of the most prominent application uses and current research at LEO is presented in Table 2.

Table 2 Application domains and state-of-the-art research in LEO satellite communications.

Reference(s)	Application	
Campbell, Melebari & Moghaddam (2020)	Terrain and topography monitoring using Global Navigation Satellite System Reflectometry (GNSS-R)	
Huang et al. (2019)	Landslide monitoring for early detection and disaster recovery applications	
Anghel et al. (2019)	Monitoring large built structures using satellite interferometry	
Chen et al. (2020b)	Urban densification analysis using satellite imagery	
Mendez-Espinosa et al. (2020)	Analysis of air quality during pandemics and periods of extended lockdowns	
Ouaadi et al. (2020)	Monitoring wheat crops using satellite interferometry	
Castaño et al. (2020)	Disaster prevention scheme to monitor thermal anomalies and volcanic eruptions using infrared satellite imagery	
Chen et al., 2020a	Monitoring regional development by analysing land use and land cover areas from satellite imagery	
Rauste et al. (2012)	Landslide and soil stability monitoring using satellite imagery	
Gawuc et al. (2020)	Climate change and urban heat islanding effects monitoring based on MODIS spectroradiometer (Minnett, 2019)	
Xu et al. (2020)	Analysing global vegetation drought from remotely sensed data using MODIS spectroradiometer	
Jing et al. (2020)	Analysis of natural calamity such as earthquake using passive microwave radiometry	
Sturdivant & Chong (2016)	Satellite link and IoT connectivity	
Palma & Birkeland (2018)	Arctic terrain monitoring IoT system by deploying free swamp satellite constellation	
Minei & Cohen (1999)	High-speed internet delivery through satellite links	
Gavrilă et al. (2020)	Software-defined radio-based satellite gateway for IoT applications extending 5G and beyond systems with IoT edge on satellites	

Since the 90s, the fast-paced satellite deployments mostly favored terrain and climate monitoring. Currently, with the advent of micro-satellites where a constellation of free-floating satellites to cover a broader range in LEO is a common deployment strategy for both cost-effectiveness and operational ease. As reported in Table 2, an increasing trend in satellite-based service delivery models, such as internet access, backhaul connectivity access, gateway connectivity, and software-defined radios and their interconnection to ground-based IoT systems is gaining momentum. This service delivery model emphasizes the need for low-cost micro-satellites to deliver high bandwidth, high data rate services to connect with ground systems. The research in this domain primarily focuses on deploying these micro-satellites in Lower Earth Orbit (LEO) as different orbital ranges and altitudes provide different coverage patterns and are also associated with varying delays based on the distances (Mukherjee & Ramamurthy, 2012). However, this service delivery model is prone to time synchronization issues that may cause a communication bottleneck. Many researchers argued that on-board satellite time synchronization is essential to the autonomous operation of the satellite cluster (Chunhao et al., 2013; Gu et al., 2015; Ma et al., 2019). To this end, Jiuling et al. (2018) proposed a time synchronization based technique to automatically compensate for time drifts in micro-satellite clusters.

In future IoST networks, we may see a trend in choosing different orbits for different service delivery networks based on IoT applications and the coverage area intended to be focused. Satellite networks are also known to provide point-to-point, broadcast, and multi-cast communication links, enabling them to be a parallel backbone to our conventional internet infrastructure. Figure 5 detailed an architectural overview of the future IoST systems, while Table 2 lists the technical details of the technologies used in this architectural concept. It is well defined that IoST initially demands large bandwidth. High data rate enabled communication links that might enable hybrid band micro-satellites in LEO to support narrow and wide-bands. Table 3 listed some of the commercially available satellite solutions for LEO communication. At this stage, it is essential to question why companies and research institutes are not focusing on conventional satellites? (William, 2010). Is the service delivery model going to blend with the IoT service models? In the IoST domain, the significant technological challenges are based on hardware; the micro-satellites themselves. Conventional satellites cost millions of dollars, and it takes years from design to deployment. On the other hand, micro-satellites are smaller in size, cost-effective, easy to be deployed in constellations that can expand or shrink as per needs, and easily launchable (You, 2017; Lee, 2016). Some of these conventional systems’ limitations are the lack of continuous coverage, lower data rates, using conventional satellite frequency bands that are already very congested, and high operational and maintenance costs.

Figure 5 Internet of Space Things (IoST) system architecture concept (Akyildiz & Kak, 2019).

Table 3 Existing satellite-based broadband networks (Akyildiz & Kak (2019)).

Parameters	Iridium NEXT sensor POD	TinTin	Astrocast	Fleet	KIPP	
Used CubeSat	Yes	No	Yes	Yes	Yes	
Company	Iridium Comm, USA	SpaceX, USA	ELSE, Switzerland	Fleet, Australia	Kepler, Canada	
Purpose	Sensing & Communication	Broadband Network	IoT & M2M	IoT	Satellite Backhaul	
ISL Capability	Only to host satellites	Yes	Yes	N/A	N/A	
Deployment Year	2015	2015 (trials)	2018	2018	2018	
Orbital Altitude	780 Km	340 & 1,200 Km	N/A	580 Km	N/A	
No. of Satellites	66	7518 & 4425	64	100	140	
Frequency Band	L & Ka	V, Ka, Ku	L	Ka	Ku	

As presented in Fig. 5, the proposed IoST ecosystem will be equipped with active sensing, passive sensing, IoST hub, and a mesh of Newly designed CubeSat satellites (Saeed et al., 2020). CubeSat are low-cost, small-size satellites with customizable payloads, offering high temporal and spatial data capture facilities (Loisel, 2021). CubeSat satellites’ control and data planes are separated by incorporating software-defined network and network function virtualization. CubeSat is one such micro-satellite that is being deployed in many latest research projects (Wu et al., 2020; Piñeros, Dos Santos & Prado, 2021; Cappelletti & Robson, 2021; Kovar et al., 2020). Weighing only close to 1.5 Kilograms and having a cubic dimension of 10cm, CubeSats can be deployed by Poly-Pico Satellite Orbital Deployer (PPOD) System (Chin et al., 2008).

Akyildiz et al. proposed a new CubeSat with reconfigurable multi-band radios for satellite communications. Their research on using THz frequencies for these satellite links and the use of graphene-based antennas on these high frequencies show a potential application for high bandwidth and high data rate requirement components (Akyildiz & Kak, 2019). One of the most prominent features of their proposed model is to equip the CubeSat devices with IoT-like sensors that can perform passive sensing for several applications. These sensors may include a high-resolution camera on the micro-satellites that enables high-resolution imagery for an IoT application running on the ground networks. Table 4 listed some of these application-specific scenarios where these micro-satellites can be utilized to their full potential.

Table 4 IoST ecosystem use-cases and potential applications.

Use-cases	Applications	
Backup network in the sky	Remote Internet Connectivity
Disaster Recovery Backup Network (Das et al., 2019)
Backhaul capacity additive network	
Terrain monitoring	Environmental changes monitoring
Geofencing and border monitoring
Continous object tracking	
Collaborative-aware real-time interface	Realtime intelligent transportation
Always-Connect info link for autonomous driving
Data aggregation for smart cities	
Unmanned aerial vehicular operations	Extended coverage
Real-time trajectory correction
Real-time flight path correction
UAV network in the sky	

Remarks

IoST may extend our current IoT and 5G solutions for high bandwidth, low-latency applications. The idea of utilizing scalable micro-satellite constellations for enhanced network coverage and on-demand backhaul network are some of the very promising aspects. However, the integration of micro-satellites with current IoT infrastructure, poses a lot of challenges. Some of the major challenges in this domain include resource allocation, long inherent transmission delays, complex network topology and inter constellation convergence. The resource allocation is a very critical task that requires a dynamic scalable solution to optimize available bandwidth, transmission power, resource pooling and scheduling. In addition, this requires optimizing physical and link-layer parameters per application-basis, which may further complicate the task. Therefore, a trade-off between service availability and resource optimization is inevitable.

Furthermore, IoST networks must be built by deploying constellation of micro-satellites that requires strict time synchronization. This requires dynamic scheduling and routing protocols for time synchronization and fast network convergence. Software defined radios (SDR) may help to resolve this issue in the future by separating control and management planes of the network. In addition, network function virtualization (NFV) can be adopted for inter-constellation synchronization and resource scheduling. Finally, the use of optical waves to establish inter-satellite links can ensure a reliable communication link for realtime as well as backhaul network applications.

Internet of Underwater-things (IoUT)

Oceans cover more than 70% of the Earth’s surface and about 97% of the Earth’s water supply. Ocean currents cycle is the main factor that controls the world’s weather (Nunez, 2019). It is interesting to know that mega projects like Google’s Project Loon (Salinas, 2018) and Facebook’s Aquila (Heath, 2015) connect remote areas of the Earth by providing internet coverage, thus accounting for only 3% of the surface that makes this planet. Whereas, the vast ocean floor is left untouched to this day, which is why it is very natural for scientists and researchers to deeply study the ocean bed and explore possibilities to interconnect devices and sensors. The underwater network can collaboratively build up a system of IoUT, which is one of the most active and open research areas currently.

The bigger goal to investigate the possibility of IoUT is to build a system of interconnected underwater devices, sensors, and autonomous vehicles that can communicate and relay information via the internet. The network of underwater things will enable scientists to deploy disaster recovery and identification solutions such as oil spillage, tsunamis, prominent shipwrecks, and bio-sensors to detect ocean bed quality and study coral reef the health of deep ocean species. Underwater networks may also provide a substantial advantage for military applications by ocean geo-fencing, which could become the tool of choice for international waters monitoring and administration.

It is essential to consider that the standard communication technologies that operate on land do not work underwater, which proposes a more significant challenge to build sustainable technology that can scale and be plugged into the internet to transfer the vast amount of underwater sensed information. However, radio waves can communicate underwater, with lower frequencies, bandwidth, attenuation, and considerable propagation delays. These drawbacks limit their operation to only a few meters in shallow sea applications (Palmeiro et al., 2011; Jahanbakht et al., 2021). Figure 6 presents a reference architecture model of Underwater Wireless Sensor Networks (UWSNs) that will help us to understand the technological challenges at various layers. To accomplish the more extensive IoUT roadmap, these underwater sensor networks’ deployments must be first understood and standardized and investigated deeply into the underlying technology that can support high bandwidth high data-rates and connect to satellite and IP-based networks. A detailed application scenario for IoT based underwater communications is presented in Fig. 7.

Figure 6 A reference underwater sensor network architecture.

Figure 7 The Internet of Underwater-Things real-world application scenarios.

Laser-based underwater communications were also investigated as the proposed deep-sea communication technology (Wu et al., 2017; Schirripa Spagnolo, Cozzella & Leccese, 2020; Nguyen, Nguyen & Mai, 2020; Zhu et al., 2020; Ramavath, Udupi & Krishnan, 2020). However, due to the varying nature of ocean tides, biological materials, waves, and turbulence limit such technology’s use over only short ranges. Some of the current work on underwater optical communications is based on high frequency, narrow-bandwidth blue laser waves that exhibit lower attenuation and can provide data rates up to several tens of Mbps (Palmeiro et al., 2011; Nguyen, Nguyen & Mai, 2020; Jaffe, 2014). Another aspect that strictly limits cutting-edge underwater technology is the increased cost of implementing such sensors, undersea links, and vehicles (Shantaram et al., 2005).

To this end, it is argued that optical communications may provide high-reliability, secure underwater communication networks. However, the cost of deployment, technical complexity and power consumption exponentially increases (Li et al., 2019; Khalil et al., 2020). Such technological issues push a great deal of underwater communication research back towards acoustic technology, whereby varying frequencies of multiple tens of kilometers area can be covered. However, there are severe degrading effects of multi-path reflections in acoustic technology dependent on the oceans’ depth and ever-changing rugged landscape (Heidemann et al., 2006; Sozer, Stojanovic & Proakis, 2000). However, it should be noted that in the future, IoUT will adopt acoustic communications by deploying billions of underwater sensors and vehicles. The multi-path reflection effect from these devices may severely jargon communication. Some of the significant differences between Traditional Wireless Sensor Networks (TWSNs) and UWSNs are presented in Table 5 (Kao et al., 2017; Yusof & Kabir, 2012). Erol-Kantarci, Mouftah & Oktug (2011) in their survey on Acoustics-based underwater sensor networks, identified a very distinct differentiation in the architectures. These localization techniques are needed to build up underwater sensor networks. They presented an exciting approach to group various underwater acoustic sensor networks (UASNs) based on a centralized versus decentralized nature, arguing that decentralized architecture requires local processing on the node. Whether estimation or prediction-based, localization is vital in these sensor networks is crucial as the underlying ocean bed is always abruptly changing. However, it was observed that performance analysis on these underwater localization techniques, location-based, and cluster-based routing techniques and their performance is still unrecovered and stands as an open challenge for underwater communications.

Table 5 The differences between TWSNs and UWSNs.

Features	TWSNs	UWSNs	
Transmission media	Radio wave	Sound wave	
Propagation speed	300,000,000 m/s	1500 m/s	
Transmission range	10–100 m	100–10,000 m	
Transmission rate	2~50 kbps	1~0 kbps	
Difficulty to recharge	Depends on the application	Difficult	
Node mobility	Depends on the application	High	
Link reliability	Depends on the application	Low	

Song, Stojanovic & Chitre (2019) in their research article, presented the current standings in underwater acoustic communications, namely the merging applications, challenges in network stacks, and the implementation issues. A significant increase in network-based schemes such as localization, routing, multi-hop communications was observed during the last decade. However, there has been less previous research on application-specific architectures. The signal processing techniques for communication channels have a varying effect as acoustics communications are broadband, and with an unpredictable underlying channel, synchronization still stands a considerable challenge. Some of the primary vital issues in implementation revolves around reliable communication and networking between these devices. Some of the top research initiations are made by Project SUNRISE, which focuses on underwater robots and autonomous vehicles capable of communicating with each other and transmitting information via the internet. Inter-robotics communication is based on acoustic signaling. A great deal of research is being carried out to address the unpredictable underwater temporal variations that make acoustic communication difficult. Researchers are also looking into adaptive acoustic signaling to account for the underwater environment (Sunrise, 2017).

The underwater communications suffer from propagation delays, multi-path fading as well limited bandwidth. These challenges require adaptive MAC and routing protocols to overcome some of the associated limitations in underwater environments. Lee et al. (2013) presented a comprehensive review of MAC protocols for underwater communications. The proposed fusion architecture achieved improved packet delivery ratios to overcome multi-path fading and mobility challenges. Similarly, Di Valerio et al. (2016) presented a self-adaptive protocol stack for underwater sensor networks to address some of the current research challenges in underwater communications. Multiple MAC protocols were chosen for the best-fit criteria. It was discussed how these protocols were dynamically selected according to the change in network conditions. They implemented the network stack as a Software Defined Communication Stack (SDCS). They analyzed a policy engine that enabled selecting the most optimal MAC protocol based on network conditions such as network conditions, configurations, and different packet payloads.

The experimental results are encouraging as they gave a proof of concept to implement Software-defined stacks with the ability to have a self-adaptive communication channel. However, these results were carried out in the controlled environment, and an actual underwater communication channel also poses sever channel variations that can affect the transmission medium. The policy engine may struggle to select the best-fit MAC protocol to achieve the desired throughput.

Another research carried out under the same project made use of the OptoCOMM modem (Caiti et al., 2016a) and SUNSET Software Defined Communication Stack (S-SDCS) to overcome the data-rate and bandwidth limitations of acoustic technology. A large chunk of files summing up to Gigabytes was transmitted using these optical underwater wireless modems (Caiti et al., 2016b). The results showed the capability to transfer such large volumes of data wireless. It was also observed that the software stack provided the capability to offload such high and arbitrary file-sizes successfully. However, the implementation details and the dynamics of the underwater communication channel can severely affect communication. It is promising to see the optical modems installed on the gateway nodes with the capability to offload extensive data from these underwater sensor networks.

Remarks

IoUT is an emerging ecosystem that aims to connect underwater smart objects with other maritime resources. This fast-emerging ecosystem may facilitate in navigation, positioning, underwater exploration and disaster prevention across the world. Inheriting its DNA from IoT networks, the technology layer of IoUT networks is largely heterogenous. However, the communication technologies are still very limited, mostly utilizing acoustic and optical waves. Existing underwater communication mostly utilize acoustic waves for long distance communications. However, these systems have two major drawbacks. First, these technologies cannot be used for security and monitoring applications as they lack stealth and can easily be intercepted by adversaries (Yisa et al., 2021). Second, these systems have very small bandwidth, thus supports very low data-rate applications. In recent years, optical waves have been utilized to overcome the underwater bandwidth challenge, however optical communications has its own challenges. Optical communication systems suffer in transmission range, as well as they are highly directional. The highly directional beamforming also suffers multi-path fading and distortion resulting in synchronization and link failure issues.

Latency is an important consideration when selecting communication medium. Underwater communications suffer from high latency that are in some cases hundreds of times higher than terrestrial networks. However, scientists are investigating optical modems to reduce the latency for underwater communications.

Another critical challenge stems from the mobility of underwater devices. Due to the dynamic nature of underwater environments, the network topology is never consistent. This requires a thorough investigation into fast converging routing protocols and self-healing underwater networks. Finally, the rugged underwater environment also influences the network and device lifetime. Underwater devices are mostly battery powered and have a limited lifetime. Unlike conventional terrestrial devices, there batteries cannot be charged or replaced frequently. In addition, conventional renewable energy sources such as solar, cannot be used underwater. Therefore, new energy harvesting techniques must be deployed that can generate electrical energy from ocean currents. Furthermore, energy conserving routing and scheduling protocols must be designed to further conserve the battery life of underwater devices.

Social Internet of Things (SIoT)

IoT networks initially aimed at pervasively connection billions of things to interact with humans, machines, and smart objects for information gathering, control, and actuation. The pervasive interactions between machine-to-machine (M2M) communications paved the way towards truly ubiquitous computing (UC), that is intelligent and able to make decisions without human intervention. Figure 8 illustrates this principle to achieve UC. This requires a direct connection between humans and machines for future networks. However, human interactions are mostly defined by relationships, trust, and networks that are need-based. Therefore, it is crucial to understand that a true human-machine interaction would require an improved connectivity between the users and the things. In addition, it requires a direct connection of things as a social connection to humans, thus, evolving as ubiquitous social internet of things (SIoT). This contextual extension of information flow between humans and things will enable smart IoT networks that are data-driven and operate in relation to social human interactions. The social driven IoT networks will not only increase quality of experience (QoE) for humans but will also introduce ambient intelligence to the systems (Ortiz et al., 2014).

Figure 8 Human and device interaction towards pervasive SIoT (Ortiz et al., 2014).

The Social Internet of Things (SIoT) connects social networks and IoT. The core concept is that a large number of people linked together in a social network may deliver significantly more accurate answers to complicated issues than a single individual. Thus, one essential goal in better implementing services inside a specific social network of objects will be to publish information/services, identify them, and discover new resources. This can be accomplished by perusing a social network of "friend" things rather than depending on traditional Internet discovery techniques that are incapable of scaling to billions of future devices.

SIoT is centered on fundamental connections such as the parental object relationship (POR), which is created between items in the same product category, and the ownership object relationship (OOR), which is based on heterogeneous items belonging to the same user (For example, mobile phones, game consoles). Such interactions should be formed and managed without the need for human intervention. Humans are solely in charge of establishing the rules for things and their social interactions. SIoT, in a nutshell, draws a connection between today’s social networks and a future network of things (Romdhani, 2017).

SIoT paradigms

The social integration with internet of things (IoT) requires stringent implementation actions that could extend from the existing IoT solutions. IoT at its core is a conjunction of heterogenous devices and enterprise services that lacks standardization and trust boundary compliance. It is therefore crucial to understand the significance of various SIoT perspectives that are needed to understand the device as well as human behavior in this ecosystem. In this article, the SIoT implementation is investigated based on the following perspectives.

Human–device interaction

As explained in Fig. 8, the IoT integration with social networks requires communication between devices as well as human interaction. This involves device-to-device (D2D), human-to-human (H2H), and human-to-device (H2D) communications. At present, the IoT technological landscape provides pervasive D2D communications. However, H2D integration still requires improved architectures and middleware for seamless integration. This interaction is required to achieve ambient intelligence based on H2H interactions, thus bringing a step closer towards truly ubiquitous IoT networks.

Collaborative-awareness

The collaborative-awareness, particularly in SIoT networks is an important consideration in contrast to conventional IoT networks. The traditional IoT networks operate on reactive and pro-active data from devices that gather actionable intelligence to perform various tasks. On the other hand, the SIoT ecosystem will require the need based H2H interactions, that will regulate D2D, and H2D communication. SIoT networks must appear as an extension to social networks to increase collaboration, and quality of experience (QOE) for humans (Ortiz et al., 2014).

Data protection and privacy

SIoT environment is an example of pervasive computing based on need-based human data. Typically, devices including (sensors, mobile phones, passive internet devices, etc.) are used for data acquisition, that may take many forms. In broad context, SIoT networks must regulate the reactive and pro-active nature of the data. This is crucial to maintain the data protection and privacy in the entire process, that is, data in motion, as well as data at rest. The data acquisition can invoke data-crawling and learning methods to pro-actively analyze the situation-based learning. Whereas the reactive data acquisition may require a real-time query. Therefore, in SIoT context, data acquisition must comply with security, data protection and privacy compliance.

In order to establish SIoT as an ecosystem, these holistic perspectives must be studied in the context of a framework. This will allow IoT enabled smart devices to interact with humans, social networks following an architectural neutral framework, that could integrate current enterprise solutions, applications and various IoT based services to social networks. The SIoT framework requires an insight into the fundamental building blocks, that are given in Fig. 9.

Figure 9 Social Intenet of Things fundamental building blocks.

It is critical to comprehend these crucial building blocks, which are outlined below.

Social network integration

The integration of smart objects (SO) into existing (SN) is very important to facilitate the data acquisition onto which ambient intelligence could be generated. This requires the service discovery, operability within user trust roles (TR), and service connections to SNs. This can be achieved by integration of SN APIs to fetch reactive data that could be shared with SO for informed decisions. An important consideration at this stage requires the user roles and permissions compliance as regulated by the SNs to ensure data integrity and SN to SO relationship. By extending the SNs data updates, such as status, location, and work updates can be used to control SO for SIoT networks.

Ambient intelligence

Ambient intelligence in SIoT can only be derived by interfacing devices to SN dynamically. Typically, in UC, the devices will be controlled automatically by human actions on their SNs. This requires regulation to maintain a consistent relationship between devices and humans. In addition, the notion of intelligence is required for D2D communications that follows the service protocols of SNs. Some studies suggest the use of middleware components to regulate this trust management and data flow policies of SIoT networks.

Social networks and smart objects collaboration

SN and SO collaboration is at the heart of the SIoT technology landscape, as it dictates the communication between devices and SNs. Most of the currently available IoT devices are based on embedded platforms that wirelessly communicate with each other and gateway devices to exchange information (Ali, Ishak & Bhatti, 2021a). These embedded devices are constrained in nature and requires efficient protocols to communication with each other. On the other hand, the SNs are mostly cloud-based enterprise solutions with improved computational and memory resources. As a result, SIoT architectures may require ground-up modeling in order to integrate with SNs efficiently. These architectures are later discussed in the following section.

Everything-as-a-Service (XaaS)

The cloud computing “as-a-Service” concept, which provides services through the internet, is a global trend that is gaining traction. SaaS (Software-as-a-Service) options are quickly becoming the de facto way for customers to access services, products, and enterprise solutions. Other types of services are being made accessible under the same pay-as-you-go business model, including Platform-as-a-Service (Paas), and Infrastructure-as-a-Service (IaaS). The XaaS model can be thought as the integration of all these cloud business models. At the core of SIoT has been the concept of transforming things and SNs into services and of enabling them to be readily discovered and combined with various other services. Using IoT devices and analytics to improve value proposition and QoE is something that will be seen in future SIoT networks.

The SIoT taxonomy

The SIoT taxonomy is now presented based on the aforementioned paradigms and building blocks. The SIoT taxonomy as illustrated by Fig. 10 is used to investigate the literature in this domain.

Figure 10 The Social Internet of Things (SIoT) taxonomy.

Although various IoT architectures have been proposed in the past, there still lacks a complete architecture for SIoT domains. Most IoT architectures are based on layered approach that abstracts the hardware, network, and applications. Due to the heterogenous nature of IoT devices, underlying technology, network stack and end enterprise implementations, there lacks a standardized architecture in this domain (Al-Fuqaha et al., 2015). In recent years, middlewares have been proposed to abstract the communication and network layers to interface between smart devices and enterprise applications/services. Ideally, SIoT architectures must support layered architecture with components and service abstractions managed by middlewares. In addition, the middlewares can also enable the trust management and privacy governance while integrating SO to SNs. Table 6 compares several proposed SIoT architectures in detail.

Table 6 Various proposed architecture models for SIoT domain.

Year	Architecture type	Description	Strengths	Weaknesses	Reference	
2010	Relational	Real-world implementation of hierarchical relational model based on smart objects	Policy-aware model that can be extended to SIoT domain	Model based on RFID technology that may not be directly integrated to smart devices and social networks	Kortuem et al. (2009)	
2011	Relational	Smart objects based relational architecture	Interfacing humans in the loop with smart objects	A theoretical concept	Atzori, Iera & Morabito (2011)	
2014	Generic	A generalized architecture to connect smart objects to Social Networks	Addresses the data, privacy, and interface needs for SIoT networks	A theoretical concept	Ortiz et al. (2014)	
2016	Layered	A client-server network inspired architecture for SIoT	Model can be extended to abstract layered functionalities	No clear directions for implementation	Tripathy, Dutta & Tazivazvino (2016)	
2017	Relational	Smart Objects based architecture for SIoT	The ontology-based model follows object-oriented approach and can be implemented easily	The study relies on quantitative reasoning. The implementation details were not thoroughly documented	Kim, Fan & Mosse (2017)	
2019	Layered	A semantics-based layer architecture for SIoT	N/A	A theoretical concept	Gulati & Kaur (2019)	

The most significant goal is to realize a social network of smart things that can interact naturally with each other by understanding each other’s trust and privacy requirements to form virtual groups. These virtual groups or societies are scalable, share information and resources, inference on each other’s network-based intelligence, and collectively perform a task if required. Kranz, Roalter & Michahelles (2010) first presented the idea of realizing future pervasive computing where human users and technical systems collaboratively provide service to each other. They demonstrated a cognitive-aware office model where everyday things were made smart using RFID.

Atzori, Iera & Morabito (2011) first referred to this concept as a unique evolution of “Smart things” to “Social Things”. They put forward a concept where devices or, in this case, referred to as simply objects form different types of associations that are based on co-parental, co-work, a co-objective, or social grouping of one or all the above. The researchers believe a handful of smart objects, including mobile devices, RFID fobs, smartwatches, from multiple vendors capable of performing various tasks (similar or completely different), can be put to work in the same location. This strategy will allow a human-like social gathering of smart objects which is sometimes location-based, continuous or sporadic. This social object relationship model with similar capabilities forms the basis of social network and could be compared to the human social networking model given in Fig. 11.

Figure 11 Basic components of social networks models of human versus objects.

Moreover, Atzori et al. (2012) further extended their social object framework concept to outline the overall concept and architecture of SIoT by characterizing the underlying networks, objects, and functional requirements. Their idea revolved around a flexible network with virtually no boundary and a hierarchical level of trustworthiness that governs the relationship between these objects. The researchers also claimed that humans’ interaction patterns could be directly applied to objects in a pervasive environment where these behaviours or patterns can be mapped and programmed to their respective services or actions. A four-layered service model was also proposed to model these behaviours from the early stages of device provisioning, service discovery, and relationship to service provisioning. This research was a comprehensive way forward where the application examples highlighted the need and future adaptation of SIoT networks.

Ortiz et al. (2014) investigated the integration of human interactions with IoT networks in order to create social conscience-based networks. They built a user-roles-based relationship between smart objects and social networks in the suggested research, which may also infer SN-based intelligence for communication. Furthermore, the proposed architecture included a consumer-product connection based on trust to facilitate D2D, H2H, and H2D interactions. This study is an important step in decoupling the fundamental building blocks, user roles, and integration strategies for future SIoT networks. However, no precise implementation strategy was offered, which may have established an appropriate platform for implementation.

Kim, Fan & Mosse (2017) presented a client-server architecture for SIoT applications . The three-tiered architecture is built on a client application for D2D communication, a server component for human and SN interaction, and databases for accessing user data. The client application begins device access and access rules to manage end-user communication, whereas the server client provides general access among all components. This architecture may appear to be suitable from the standpoint of communication; nevertheless, user roles, trust, and privacy must rigorously conform with the SNs and data rules of the relevant network. Furthermore, this approach lacked a defined plan for scaling and integrating with multiple devices and networks.

Ganti, Ye & Lei (2011) proposed the concept of trustworthy sensing-enabled devices that collect and exchange data for communal processing and measurement of pertinent information. Smart objects such as GPS sensors and mobile phones were employed in their examples to collectively exchange data, which was then aggregated and displayed to make informed decisions. The study shed light on the worth of data supplied by items. Because the majority of the data flowing from these nodes contains user or private data, it alluded to the need for a trust, privacy, and governance paradigm.

Gulati & Kaur (2019) proposed a four-layer semantic-oriented platform architecture for SIoT as a reference model, with four layers: objects, communication, SIoT management, and application. The lowest layer is the layer of objects, which refers to the objects that are placed on the SIoT and communicate via local networks and sensors. Protocols, gateways, and technologies needed to communicate amongst objects are included in the communication layer. The SIoT management layer relates to SIoT platforms and services that comprise key components such as identification management, object profile, proprietary control, discovery of services, composition of services, trust management and relationship management that communicate through the API to the app layer. Although a thorough SIoT architecture is presented, which allows information flow at every layer, implementation details and interface with current solutions and platforms are still lacking.

Kortuem et al. (2009), on the other hand, concentrated on the categorization of privacy-based characteristics for SIoT designs. In their study, smart objects were characterised by their activity, process, and policy-awareness, which allows for the deployment of smart objects based on trust and privacy needs in various application situations. This categorization provided information on the behaviours, context, process, and regulations that control a specific device in a given application situation. It is extremely probable that these smart devices will create virtual groupings that are either:Activity-aware to perform a specific task.

Policy-aware to follow domain-based/role-based rules to accumulate data over a long period.

Process-aware and to perform context-aware tasks (which will also become a behavioral quality of future SIoT).

The strict data privacy, policy-awareness and trust compliance requires a thorough investigation of trust roles and trust management (TM) strategies for real-world applications. This trust-based relationship is investigated in the following section.

Trust roles and trust management

The challenges of trust are among the most important in present IoT technology which address the interaction between objects. Lack of confidence in things that collaborate socially causes issues such as loss of privacy, safety, security, access, and information manipulation by unauthorized people or objects (Kowshalya & Valarmathi, 2017; Nitti, Girau & Atzori, 2013). Every object in the SIoT technology stacks must have a high confidence level in its adjacent objects in order to have a trustworthy communication. A trust-based connection also facilitates to identify trustworthy and malicious objects based on the confidence levels.

Data privacy and confidentiality are one of the most critical design challenges for SIOT TM architecture. It is important to consider the heterogenous nature of these objects, that inherits a host of vulnerabilities and security shortcoming at object level. A security strategy is therefore necessary for limiting network access from assaults in the access control system. Therefore, it is crucial to design an intelligent TM control to prevent unauthorized access to objects and its associated data within SIoT networks. The identification of trust types between smart objects holds a significant importance in designing TM solutions. Table 7 presents a few trust type relationships reported in the literature.

Table 7 Trust type relationships in SIoT domain.

Trust Type	Description	Articles	
Local	The trust relationship is formed between adjacent objects directly connected to each other. Objects in extensions do not inherit the trust relationship of its parent nodes.	(Kowshalya & Valarmathi, 2017; Roopa et al., 2019)	
Direct	The direction trust relationship is only formed between two directly connected objects in the hierarchy	(Kurniawan & Kyas, 2015; Moradi & Ahmadian, 2015)	
Indirect	The indirect trust relationship may inherit roles and policies from adjacent objects based on user roles recommendations and confidence level reputations.	(Kowshalya & Valarmathi, 2017; Nitti, 2014)	
Transitive	The transitive relationships are formed on object roles. If certain objects perform a similar operation in the network, they may inherit the user-roles and trust relationships automatically.	(Mukamakuza, Sacharidis & Werthner, 2018; Gohari, Aliee & Haghighi, 2020)	
Asymmetric	Asymmetric trust relationship is formed on the confidence level between objects. Consider two objects formed a transitive relationship based on common operations/tasks, the asymmetric relationship between these objects may still differ based on the confidence level. Asymmetric trust relationships may help to further define fine-grained policy controls for network objects.	(Janas & Oljemark, 2021; Faqihi, Ramakrishnan & Mavaluru, 2020; Son, Choi & Choi, 2020)	
Context-dependent	Context-dependent relationships are temporary and are automatically formed based on objects interactions, tasks, and user-requirements.	(Roopa et al., 2019; Xiao, Sidhu & Christianson, 2015; Parvin, Moradi & Esmaeili, 2019)	
Dynamic	Dynamic trust relationships are governed by features that may change over time, or per application basis. For instance, two mobile devices may form a context-dependent trust relationship for location sharing scenario but may be deemed untrustworthy in social status updates scenarios.	(Roopa et al., 2019; Chen, Guo & Bao, 2014; Chen, Bao & Guo, 2015)	

According to the literature, trust types differ depending on the application and context. It is extremely unlikely that a single trust type profile could cover all object-specific tasks and confidence levels. As a result, an intelligent and dynamic TM architecture that adjusts to the deployment scenario is required. However, TM architecture, which correlates to trust, privacy, and data security, remains one of the most critical open issues in the SIoT domain. These research questions, as well as possible solutions, are explored in the section that follows.

Open trust management challenges in SIoT domain

The integration of smart objects, humans, and the SNs poses data leakage and security risks. In addition, it is imperative to ensure the trust management and confidentiality to comply with the data protection policies. Unlike other IoT domains where security policies are mostly devised for intra-network communications, SIoT ecosystem focus mainly on data protection and privacy. In this regard, some of the most critical open research challenges in SIoT domain include:Trust, Privacy and Security.

Trustworthiness Management.

Context-aware collaborative computing.

Much research has been undertaken on trustworthiness management in social and peer-to-peer (P2P) networks, with the vast majority centered on the concepts of reputation-based methods maintained either locally, centrally, or disseminated throughout the network. As a result, a computation model is required that quantifies a probabilistic estimation average, which aids in determining if a participant’s reputation has increased or decreased as a result of malicious information sharing with its peers (Golbeck, 2005; Nitti et al., 2017; Sun et al., 2019).

Nitti, Girau & Atzori (2013) and Nitti et al. (2017) investigated trustworthiness management in SIoT by thoroughly researching two approaches for developing scalable, dependable social behaviour in smart devices. The study focuses on supervisory control, which first allows the system to provide a certain service to an object with which it interacts. The instances of allowing or denying requests then aid in the learning and training of the model to develop direct or indirect interactions based on the requester and its characteristics (scored based on its actions and intentions). The subjective trustworthiness model focuses on the social viewpoint, how humans interact with their friends, and how they maintain the relationship level structure. The trustworthiness is based on the same principles as word-of-mouth in a human interaction setting, resulting in a trustworthy relationship that follows the hierarchy of friends or friends of friends.

On the other hand, the proposed model substantially resembled the P2P method. Each node and its related action information were saved in a Distributed Hash Table, which is accessible to the entire network but can only be managed by Pre-Trust Objects (PTO). In its most basic form, any node can request a PTO to inquire about another node’s reputation and trustworthiness. A credibility system prevents hostile nodes from delivering misleading or negative feedback. However, the proposed scheme could not provide a complete trustworthiness management model that could be directly applied to SIoT.

Morever, Nitti, Atzori & Cvijikj (2014) proposed a concept of social network construction for smart objects in which service discovery, network navigability, trust, and connection selection are all dependent on the relationship created between objects. According to the study, objects inherit certain human characteristics and use these as a behavioural feature while seeking for new friends. They believe that a first relationship or search for comparable service (in the context of smart objects) fits the sociological model.

In another study, Zhang et al. (2018) proposed a contract-based access control framework for managing access control provision for IoT networks. The distributed blockchain-based access control mechanism overcomes the single point of failures and provides authorization and validation for subject-object pairs. The proposed scheme enhances network trustworthiness by blocking access to all rogue IoT devices. In addition, the distributed architecture can scale seamlessly in an SIoT deployment scenario. Furthermore, the dynamic access policy assignments can allow for fine-grained controls per application or device-roles basis. Similarly, an attribute-based access control framework was proposed for smart cities environment, that could be implemented in SIoT domains. In Zhang et al. (2020), the researchers proposed a distributed access-control framework that combines blockchain and attribute-based access control (ABAC) model. The proposed strategy incurs less deployment costs in terms of operational complexities and can be used for SIoT environments that have a very large-scale network and data flow.

An et al. (2015) proposed a crowdsourcing task assignment model based on credible interactions between users. The proposed model investigates user interaction, relationship cognition to generate credibility ranks, including service quality factor (SQF) and link reliability factors (LRF). The rank metrics are then utilized to build a crowdsourcing community of smart objects. As shown in Fig. 12, the proposed strategy focuses on cognitive modelling of social relationships, application recommendation systems, and a credible route detection system to train the cognitive task assignment to end objects.

Figure 12 Framework—the crowdsourcing assignment model (An et al., 2015).

Along the same lines, Liu et al. (2019) proposed a real-life application scenario of SIoT and mobile crowdsourcing application for location-based networks in their study. Offline event marketing has grown in popularity among local and global, small and large-scale companies all over the world. Many models support an offline event marketing presence based on First-come-First-Served (FCFS) or Nearest First (NF). Using the concept of marketability, the researchers recommended using location-based social networks such as Facebook and FourSquare to maximize an offline event visitor’s likelihood scenario by inferring intelligence from ideally selected participants. The suggested cost-model accounted for customer loyalty, distance from the event, and the recommendation index, and it was tested to real-world datasets to demonstrate efficiency and increase in participant selection. The proposed model may adapt to a niche application scenario; however, it cannot be scaled up and implemented as a TM solution for SIoT domains.

Similarly, SIoT data security and privacy concerns are yet to be addressed. It is vital to understand the threat vectors that are linked to the entire IoT threat surface. In this regard, Rizvi et al. (2018) presented a holistic view security and data privacy aspects of IoT networks. In their research, a security taxonomy is presented that aligns open security challenges with IoT technology and components at various architecture layers. The article first identifies various IoT architectures and associated threats at every layer. This is crucial to analyze the open threat surface vulnerability and associated threat vectors that could weaken the entire security for IoT systems. In addition, the researchers defined the trustworthiness of the IoT systems by investigating overall system privacy, data availability and reliability. Finally, the importance of compliance and governance establishes the need for security practise standardisation, which could aid in risk mitigation and network security, as well as data privacy in IoT networks.

Along the same veins, in Rizvi et al. (2020a) the researchers investigated the attack surfaces by decoupling into various zones, that could help to mitigate the security risks in IoT networks. The researchers proposed a threat model to identify the system weakness and associated vulnerabilities at every layer of the architecture. This mapping of vulnerabilities is crucial in identifying the threat levels during the design, implementation, as well as integration of IoT systems. Various IoT domains including personal, health-care, transportation, and industrial IoT were studied to investigate a 1:1 mapping of device vulnerabilities to threats/vulnerabilities. The proposed model provides a visual representation of IoT network security vulnerabilities in context of trust zones and attack points.

Similarly, in Rizvi et al. (2020b), the researchers proposed a threat modeling approach at the device level to address the security issues. The proposed work investigated various IoT domains by modelling device vulnerabilities based on NICT CVSS scores. In addition, the researchers highlighted the security control measures to mitigate security threats for IoT domains.

Ali, Ishak & Bhatti (2021b) provided a similar approach to investigate the threat vectors and threat surfaces by aligning them to various IoT architectures. The proposed technique modelled an Industrial IoT device for various attack vectors at every layer. In addition, a holistic security view was provided by zoning various components and its associate data. Finally, the threat vectors were categorized into risk metrics using NICT-CVSS scores.

These techniques are crucial in identifying the overall system weaknesses for IoT domains. In general, the security of any system is evaluated by its weakest link, therefore, threat modeling is critical in identification of threat surface landscape. The security and privacy can only be improved by strict compliance and governance, that requires ground up modelling from top to bottom, and from inside out. In addition, social IoT networks are prone to additional vulnerabilities due to the vast amount of data flowing through IoT networks. The data is particularly vulnerable to attack both at rest, and in transition. Therefore, security, privacy, trust, and compliance must be investigated in parallel to be able to govern secure future IoT networks.

Mitigating security challenges in SIoT

In addition to the security, privacy and trustworthiness challenges in this domain, the real-world implementation may still require additional insights and conclusive solutions. Some of these challenges and possible solutions are presented in Table 8.

Table 8 SIoT security, privacy, and trustworthiness challenges and proposed solutions.

Challenges	Solutions	
Security, Trust, and privacy	Trust Management Systems
User Access Control Systems
Lightweight data encryption
Block-chain based data management framework (Tong et al., 2019; Wei, Wu & Long, 2020; Yi et al., 2020)	
Device Heterogeneity	Object identification based on computational resources
Device clustering based on resource-based relationships
Resources-based policies and neighbors selection	
Interoperability	Resource based task assignment and relationship modelling to allow interoperability between heterogenous smart objects	
Mobility	Mobility management based on one-to-one location-based schemes
Proximity based sub-communities for smart objects	
Service Discovery	Artificial Neural Based models for faster service discovery and network convergence
Service advertisements to initiate role-based community formation	
Context Management	Unified semantic based context management must be employed to overcome the heterogeneity issues that supports interoperability and consistent information flow	
Application Development	Middleware based lightweight Application Programmable Interfaces (API) may help in application development and deployment.	

Security, trust and privacy

Keeping in view the heterogeneity of devices, multitude of services, security is the most important building block for SIoT ecosystems. Secure and trustworthy SIoT networks are required to safe-guard various attacks, whether network-based, access-based or data-based. This sensitive requirement will ensure reliability and resiliency in SIoT interactions. It is imperative to understand that without trustworthy and secure SIoT networks, it is more likely that SIoT could lose its potential before even gaining momentum. Therefore, the aforementioned schemes are crucial to ensure data confidentiality, platform integrity and compliance for data governance in SIoT ecosystems.

Device heterogeneity

IoT networks are mostly built on top of sensor networks that offers a range of computational and networking resources. In addition, the integration to humans (either via smartphones or computers) adds another layer of heterogeneity to the technology stack. Furthermore, the technological stack implementations including protocols also vary among these devices. Finally, due to lack of standardization, a host of IoT application solutions further increases the heterogeneity. It is therefore very important to ensure device interoperability for SIoT ecosystems. Role-based and resources-based smart objects clustering can effectively resolve this problem, whereby group of similar tasks/resources smart objects can build direct relationships. The implementation flexibility allows manufacturers to produce cost-effective IoT devices. On the contrary, the lack of standardization will hamper the effective implementation of these resources in SIoT domains. Therefore, researchers, manufacturers and technologists must work in parallel to overcome the problems caused by technology heterogeneity.

Interoperability

Technology heterogeneity may also challenge data integrity and interoperability for SIoT networks. In particular, unified data management and storage techniques will ensure intelligent interoperability. One way of looking at this problem is to model resource-based tasks using metadata heuristics to identify the nature of data, which can be further converted to actionable intelligence. In this regard, resource management and descriptive frameworks, especially ontology-based schemes can be deployed on middlewares for meaningful data exchange between smart objects.

Mobility

Smart objects that form the backbone of SIoT ecosystems, are always in motion. It is due to this dynamic mobility requirement, one of the significant challenges is to maintain service discovery and fixed relationships between devices. In addition, the devices may change their roles dynamically based on various locations, thus, effecting network convergence and fixed trust boundaries between devices. Proximity based smart objects relationships can be used to solve this issue. Dynamic, small sub-communities of smart objects can be formed based on their locations, associated services and roles. The network can fast converge by discovering adjacent objects based on the location discovery.

Service discovery

Device mobility, interoperability, and the volume of data in SIoT networks requires efficient service discovery for faster network convergence, trust relationships, and thus information flow. Unlike conventional IP-based system, the service advertisements and discovery algorithms may not scale well for SIoT networks. Therefore, artificial neural networks-based service advertisement controls can be implemented in the middlewares to initiate role-based and location-based service discoveries. These techniques will greatly reduce the service discovery times by proactively advertising the available resources based on the smart object’s locations and resource profiles.

Context management

The adaptation of technology by consumers provided flexibility in accessing information. On the other hand, it added a significant challenge for context management of information and workflow. Consider a typical use-case of an office environment where an individual uses a host of internet-connected products (including a smart watch, a smart phone, and office PC) at the same time. These devices could be used to access SN and associated information, thus causing context switching problems. The seamless context management is required to ensure a balanced information exchange between multiple devices, while at the same time managing this information for intelligent context and device switching. In this regards, semantic-based context modelling approach can greatly extend device interoperability, easy context switching and scalable application availability amongst multiple devices.

Application development

IoT networks are usually deployed for data sensing and analysis to gain actionable insights that is needed to intelligent automate the business workflow. In SN contexts, this information is mostly accessed by various applications that are tailored to user-needs, requirements, and roles. Typically, these applications are designed to be deployed either on device, or in the cloud. The variations in computational, networking and storage resources further adds to the complexities of network architecture and device interoperability. Therefore, application development and deployment can be optimized by providing lightweight APIs for unified access across various devices. In addition, the APIs can be managed through a middleware to bridge the gap between devices and enterprise solutions, thus allowing a seamless integration of resources throughout SIoT ecosystems.

Conclusion

This research article provided an in-depth examination of rapidly emerging IoT ecosystems, which differ from traditional IoT applications in terms of design and service. Despite extensive research into IoT technology and its integration with cloud enterprise solutions, the ecosystems highlighted require a complete overhaul and are under investigation. Because of specialised service-based deployment, the IoT domains mentioned in this article appear to be a complete ecosystem. There has been a significant amount of research done on Body Area Networks, Health Care environments, including patient and hospital management systems. IoNT, on the other hand, will develop as an entirely distinct application area for scenarios such as drug delivery, drug monitoring, inter and intra-cell communication, to mention a few. This side of the technological divide, which is largely in the physical and communication layers, is impeding the progress of IoNT technology. The convergence of the IoNT with the Internet of Bio-Nano Technologies IoBNT as the future in health-care contexts is intriguing. Nonetheless, despite technical advances in these specialized study areas, much more study is needed to reach its full potential.

The research also examined the practicality of IoST as a global backhaul network. IoST may bring a wide range of applications mostly involving high-speed internet and multimedia connectivity, disaster recovery backup networks and remote data backup solutions, to mention a few. However, the implementation is currently hampered by the design and deployment of low-cost, high-bandwidth capable satellites that comprise the satellite backhaul network in low-earth orbit (LEOs). The breakthroughs in cutting-edge technologies, such as THz communication, and software-defined networking stand as enabling technologies for the IoST implementation. Although steady growth in these individual domains is relatively trivial, a cohesive architecture has yet to emerge. As a result, the IoST ecosystem is one major topic that is anticipated to attract many scientific investigations.

IoUT deployment is gaining traction thanks to its environment sensing and proactive disaster recovery applications, which can aid in understanding and regulating climate and weather in the next years. The current study showed various cutting-edge technologies that are being used to develop underwater networks. In this regard, a variety of underwater communication systems as well as current research problems were highlighted. It was discovered that the most significant problem IoUT faces is based on a communication platform that can transport massive volumes of data over long distances while accounting for temporal and geographical changes in the ocean bed. These goals are still under investigation, and technologies including laser-based communications, adaptive signaling, and software-defined networking may be deployed to overcome the challenges.

Finally, the Social Internet of Things (SIoT) was discussed, which is thought to bring the "human intelligence component" to future real-time AI-powered IoT networks. The capacity to have smart-devices with near-constant internet connections, sensing and relaying capabilities, human intelligence, and SN integration makes this ecosystem the most interesting of all, since it is thought to keep "humans in the loop" for future SIoT systems. However, security, privacy, and trustworthiness are the most critical and challenging aspects that will hinder the expansion of this area and IoT systems in general. The scarcity of standards, policies, and regulatory bodies has long been a source of concern in cyber-physical systems. With the present rate of IoT expansion, unresolved vulnerabilities may pose serious threats to the overall system’s viability, reliability, and authenticity.

These ecosystems are still under investigation, and they confront technological obstacles in their respective sectors. A need for study and unresolved issues was underlined in all three scenarios, from PHY to APP layers. For instance, key technological hurdles and open research concerns in these sectors include hardware platforms, sensors, communication technologies, communication medium, and backhaul connections. The present adaption of these ecosystems is hampered by a common theme that is based mostly on communication technologies and integration with existing 5G and IoT networks. Optical communications may be the most feasible technology for use in IoNT, IoST, and IoUT domains. Furthermore, self-configuring networks with beamforming radio capabilities will ensure that optical communications technologies’ weaknesses are addressed. Furthermore, optical modems and software-based gateways may improve the design by permitting high-speed data transfer between devices. Finally, middleware-based architectures are necessary in these domains to implement data security, privacy, and regulation laws.

The authors would like to thank Dr. Hayat Dino Bedru for her extensive support during the review of this article.

Additional Information and Declarations

Competing Interests

Author Contributions

Data Availability

The authors declare that they have no competing interests.

Omer Ali conceived and designed the experiments, performed the experiments, analyzed the data, performed the computation work, prepared figures and/or tables, and approved the final draft.

Mohamad Khairi Ishak conceived and designed the experiments, analyzed the data, authored or reviewed drafts of the paper, and approved the final draft.

Muhammad Kamran Liaquat Bhatti analyzed the data, authored or reviewed drafts of the paper, and approved the final draft.

The following information was supplied regarding data availability:

This review article did not require any computer code or mathematical modeling.

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
