# Peer review of "Emerging IoT domains, current standings and open research challenges: a review"

_PeerJ Computer Science, doi:10.7717/peerj-cs.659_

## Round 0.1 · original submission · Major Revisions

Three consistent reviews have been received. The reviewers identified some merits of the paper. They also pointed that clarifications are needed for the experiment design. Please provide a detailed point-to-point response. Note that you should not cite references recommended by reviewers if they are not appropriate. Ignorance of such references will not affect our editorial decision.

Reviewer 1 ·

Basic reporting

The quality of the presentation significantly weakens this manuscript. The number of references is enough, while the importance of the proposed IoT domains and the current solutions to the proposed challenges are missing.

Experimental design

The authors summarize the research status and challenges in several IoT domains, IoNT, IoST, IoUT, and SIoT. Actually, there are many survey papers on IoT research, but few on these domains.

Validity of the findings

Actually, the topic of this paper is interesting and would attract more attention from the community. However, I am concerned that do these IoT domains really matter? The importance of these IoT domains discussed in this manuscript should be highlighted. The paper lacks a more thorough discussion of what decisive role the mentioned IoT domains will play in the future and what are the crucial effects that they will achieve.

Additional comments

The authors summarize the research status and challenges in several IoT domains, IoNT, IoST, IoUT, and SIoT. Actually, there are many survey papers on IoT research, but few on these domains. The topic of this paper is interesting and would attract more attention from the community. However, I have some comments as follows.
(1) Do these IoT domains really matter? The importance of these IoT domains discussed in this manuscript should be highlighted. The paper lacks a more thorough discussion of what decisive role the mentioned IoT domains will play in the future and what are the crucial effects that they will achieve.
(2) This manuscript should be improved by including more recent papers and high cited journals regarding the solutions of relevant challenges proposed in this manuscript such as:
(2019) Smart Contract-Based Access Control for the Internet of Things. IEEE Internet of Things Journal, vol. 6, no. 2, pp. 1594-1605.
(2019) A Hierarchical Sharding Protocol for Multi-Domain IoT Blockchains. ICC 2019, pp. 1-6.
(2021) Attribute-Based Access Control for Smart Cities: A Smart-Contract-Driven Framework. IEEE Internet of Things Journal, vol. 8, no. 8, pp. 6372-6384.
(3) I think the writing style/clarity needs more effort before it is published. Indeed, I found the paper extremely difficult to read, due not only to the poor grammar used throughout but also the unclear structure of the argument being put across. The authors are suggested to check through the paper and correct typo and grammatical errors carefully.

Reviewer 2 ·

Basic reporting

Grammar can be improved in lines 14, 15, 16 and 17.

Excerpt: "Current 5G and Edge computing advancements, set to open a new frontier for cutting-edge connectivity on IoT platforms. In recent years, a broad spectrum of communication systems, covering a variety of domains has evolved. Such a heterogenous technology integration inhibits adoption and standardization challenges."

Similarly, line 28-29 may be reworded.

It is recommended that entire article should be reviewed with the help of a technical writer to ironing out such issues.

Experimental design

Comment 1:
While the article has focussed on only on emerging domains, if a concise viewpoint is added for "established" domains, it will add value to the article. With 5G, many of the established domains will also undergo a cycle of innovation in the areas of communication and integration.

Comment 2:
In the abstract in line 22-24, I get that the paper is directed towards providing a single and sufficient source of information to the concerned researchers on various aspects of emerging IoT ecosystems. But in line 141, the article states that it addresses only the "architecture, communications, and integration aspects". I think it would be helpful if this scope is included in the abstract as well.

Comment 3:
Line 587, 588 states that "However, security, privacy, and trustworthiness are the most significant and most challenging factors that will slow down this domain’s growth and systems in general."

While the article has given due coverage to security and trustworthiness topics, it seems light on the "Privacy". Could you please add a new section addressing it, or maybe address it within the existing sections.

Validity of the findings

In the conclusion section, it will be helpful to add a future direction or future work.

Additional comments

I thank the authors for collating data from a wide set of sources on the emerging technology and presenting it in an structured manner. Please find below few comments that will help it improve it and make it useful to the readers.

Reviewer 3 ·

Basic reporting

Overall it is a well-written article - covering an important aspect of the emerging IoT system. The author's aim is to provide a single and sufficient source of information to the concerned researchers on various aspects of emerging IoT ecosystems. All sections are well covered except Open Challenges in SIoT Domain. I believe, security and data privacy is one of the major issues in the wide spread deployment of IoT in different domains. Given the importance of security and data privacy, I feel this section is way too brief. Not only it is brief, but many recent important articles are not cited in this section. Some of these papers are: 1) Securing the internet of things (IoT): A security taxonomy for IoT 2) Identifying the attack surface for IoT network, 3)Threat model for securing internet of things (IoT) network at device-level.

Experimental design

Overall, a well-designed research methodology was adopted by the authors. No changes are required.

Validity of the findings

The results/conclusions drawn based on the provided data/facts are reasonable.

Additional comments

Overall, a good read - covering many different aspects of IoT. Please extend **Open Challenges in SIoT Domain** section by including a discussion on some of the recently published papers.

---

## Round 0.2 · accepted · Accept

The paper can be accepted. Congratulations.

Reviewer 1 ·

Basic reporting

The authors summarize the research status and challenges in several IoT domains, IoNT, IoST, IoUT, and SIoT.

Experimental design

This paper meets the topics of this journal.

Validity of the findings

The topic of this paper is interesting and would attract more attention from the community.

Additional comments

The author has done a good job of revising the paper.